# Effect of Overexpression of *γ-Tocopherol Methyltransferase* on α-Tocopherol and Fatty Acid Accumulation and Tolerance to Salt Stress during Seed Germination in *Brassica napus* L.

**DOI:** 10.3390/ijms232415933

**Published:** 2022-12-14

**Authors:** Yuan Guo, Dong Li, Tiantian Liu, Meifang Liao, Yuxin Li, Weitang Zhang, Zijin Liu, Mingxun Chen

**Affiliations:** State Key Laboratory of Crop Stress Biology for Arid Areas, National Yangling Agricultural Biotechnology & Breeding Center, Shaanxi Key Laboratory of Crop Heterosis and College of Agronomy, Northwest A&F University, Yangling 712100, China

**Keywords:** *γ-TMT*, α-tocopherol, fatty acids, seed germination, salt stress

## Abstract

Rapeseed (*Brassica napus* L.) is an important oil crop and a major source of tocopherols, also known as vitamin E, in human nutrition. Enhancing the quality and composition of fatty acids (FAs) and tocopherols in seeds has long been a target for rapeseed breeding. The gene *γ-Tocopherol methyltransferase* (*γ-TMT*) encodes an enzyme catalysing the conversion of γ-tocopherol to α-tocopherol, which has the highest biological activity. However, the genetic basis of *γ-TMT* in *B. napus* seeds remains unclear. In the present study, *BnaC02.TMT.a*, one paralogue of *Brassica napus γ-TMT*, was isolated from the *B. napus* cultivar “Zhongshuang11” by nested PCR, and two homozygous transgenic overexpression lines were further characterised. Our results demonstrated that the overexpression of *BnaC02.TMT.a* mediated an increase in the α- and total tocopherol content in transgenic *B. napus* seeds. Interestingly, the FA composition was also altered in the transgenic plants; a reduction in the levels of oleic acid and an increase in the levels of linoleic acid and linolenic acid were observed. Consistently, *BnaC02.TMT.a* promoted the expression of *BnFAD2* and *BnFAD3*, which are involved in the biosynthesis of polyunsaturated fatty acids during seed development. In addition, *BnaC02.TMT.a* enhanced the tolerance to salt stress by scavenging reactive oxygen species (ROS) during seed germination in *B. napus*. Our results suggest that *BnaC02.TMT.a* could affect the tocopherol content and FA composition and play a positive role in regulating the rapeseed response to salt stress by modulating the ROS scavenging system. This study broadens our understanding of the function of the *Bnγ-TMT* gene and provides a novel strategy for genetic engineering in rapeseed breeding.

## 1. Introduction

Rapeseed (*Brassica napus* L., AACC, 2n = 38) is one of the most important sources of edible oils worldwide. It is an allopolyploid species originating from the natural hybridisation between two ancestorial species, namely, *B. rapa* (AA, 2n = 20) and *B. oleracea* (CC, 2n = 18) [1]. The nutritional characteristics of rapeseed oil are mainly determined by its fatty acid (FA) composition and vitamin E content. The major FAs present in *B. napus* seeds (canola type) are oleic acid (C18:1), linoleic acid (C18:2), and linolenic acid (C18:3), with concentrations of 61%, 21%, and 11%, respectively [2]. Among these, C18:2 and C18:3 are polyunsaturated FAs (PUFAs) that cannot be biosynthesised in the human body. These PUFAs play an important role in the protection of eyesight and the prevention of various diseases, such as cancer and cardiovascular and inflammatory diseases [3,4,5]. Vitamin E, which is present in cold-pressed oil, is an essential dietary nutritional element for all mammals, and its deficiency leads to neurological disorders, ataxia, and even death [6,7]. Increasing the quantities of FAs and vitamin E has long garnered interest in rapeseed breeding [8]. Therefore, the characterisation of key genes regulating the biosynthesis of FAs and/or vitamin E in *B. napus* will be helpful in accurately designing new germplasms or varieties of crops with high PUFA and vitamin E contents using molecular biology techniques, thereby improving human health.

Vitamin E is an essential lipid-soluble antioxidant comprising tocopherols and tocotrienols. Tocopherols are mostly present in the seeds of common oil crops and the green-coloured components of higher-order plants, whereas tocotrienols are mainly found in cereal kernels and certain tropical fruits [9]. According to the number and position of the methyl substituents on the aromatic ring, tocopherols are classified into four forms: α, β, γ, and δ [10]. Among them, α-tocopherol is beneficial to human health, as it has the highest efficacy [11]. 

In plants, tocopherol biosynthesis occurs in plastids through a combination of two main pathways (Figure 1). The shikimate pathway leads to the synthesis of an aromatic ring from homogentisic acid (HGA) and *p*-hydroxyphenylpyruvic acid (HPP) by the enzyme HPP dioxygenase (PDS1/HPPD) in the cytoplasm [12,13]. The methylerythritol phosphate (MEP) pathway leads to the generation of a polyprenyl side chain from phytyl diphosphate (PDP) in plastids [14,15]. An additional pathway for PDP production was discovered from the phytol recycling pathway [16,17]. In tocopherol biosynthesis, the two substrates, HGA and PDP, are fused together to produce the first intermediate, 2-methyl-6-phytyl-1,4-benzoquinol (MBPQ). This reaction is mediated by the enzyme HGA phytyl transferase (VTE2/HPT). MPBQ is a substrate of either tocopherol cyclase (VTE1/TC) or MPBQ methyltransferase (VTE3/MPBQ-MT). MPBQ-MT catalyses the formation of 2,3-dimethyl-6-phytyl-1,4-benzoquinol (DMPBQ) from MPBQ. TC converts MPBQ and DMPBQ to δ- and γ-tocopherols, respectively. Finally, the last key enzyme, γ-tocopherol methyltransferase (VTE4/γ-TMT), catalyses the conversion of γ- to α-tocopherol and δ- to β-tocopherol, thereby determining the type of tocopherol [18,19]. The role of the *γ-TMT* gene has been functionally characterised in *Arabidopsis thaliana*. The loss of *A. thaliana Atγ-TMT* function results in reduced levels of α-tocopherol; however, an increased γ-tocopherol content in leaves is observed, and its overexpression leads to the approximate complete conversion of γ- to α-tocopherol and an increase in vitamin E activity in *A. thaliana* seeds [20,21]. Many useful crops with a high α-tocopherol content can be developed by introducing this gene. Transgenic soybean (*Glycine max* L.) [22], lettuce (*Latuca sativa* L.) [23,24], rice (*Oryza sativa* L.) [25], and perilla (*Perilla frutescens* L.) [26] with high levels of α-tocopherol have been developed by overexpressing the *γ-TMT* gene from the model plant *A. thaliana* (*Atγ-TMT*). Recently, transgenic soybean [27] and rapeseed (*B. juncea* L. AABB, 2n = 36) [28] with increased α-tocopherol content were developed by expressing the perilla *γ-TMT* gene. Generally, tocopherols in most common oilseeds contain minor amounts of α-tocopherol and relatively higher levels of γ-tocopherol [20,29]. Therefore, improving the γ- to α-tocopherol conversion by elevating the expression level of the *γ-TMT* gene in *B. napus* seeds would be a promising strategy. 

The major tocopherol functions are related to antioxidant properties because of their ability to interact with polyunsaturated acyl groups and protect membrane lipids (especially PUFAs) from oxidative damage by scavenging lipid peroxy radicals and quenching reactive oxygen species (ROS) in photosystem II. Furthermore, they also prevent damage due to the fact of oxidation during membrane lipid peroxidation [10,30]. Tocopherols donate a hydrogen atom to lipid-free radicals, thereby neutralising the radical and terminating the autocatalytic chain reaction of lipid peroxidation and protecting cell membranes [31,32]. In plants, both tocopherol and FA biosynthesis occur in the plastid, and the tocopherol functions as an antioxidant for stabilising PUFAs against lipid oxidation in vegetable oils [32]. The balance between vitamin E and PUFA content mainly determines the susceptibility to lipid peroxidation and storage stability of rapeseed oil. However, there is a paucity of studies examining the correlation between tocopherol and FA components in rapeseed oils. A positive correlation has been demonstrated between linoleic acid (18:2) and α-tocopherol, linolenic acid (18:3) and γ-tocopherol from the vegetable oils of 14 different plant species [33]. In addition, upon studying a collection of maize hybrids, a relationship between FAs, particularly PUFAs, and tocochromanol content was observed [34]. Furthermore, the content of tocopherol and FA components in seeds from three *Brassica* oil crop species was analysed, and a significantly positive correlation was observed between α-tocopherol and the sum of C18:1 and C18:2, whereas no association was observed between the γ-tocopherol and FA components [35].

In addition to their antioxidant function, recent studies have suggested additional roles for tocopherols in diverse biological processes, including germination [36], cell signalling [37], and biotic [38] and abiotic stresses, such as those related to salt [39,40,41,42,43,44], low temperature [45,46], drought [41,47], and heavy metals [40,43,48,49]. The RNAi-mediated silencing of *γ-TMT* leads to a deficiency in α-tocopherol and elevated susceptibility to salt stress, whereas the overexpression of *γ-TMT* increases the salt and heavy metal tolerance in transgenic tobacco [39,43]. In addition, the increased sensitivity to salt stress in vitamin E-deficient mutants of *A. thaliana* has been observed. Tocopherols play a key role in salt stress tolerance not only by reducing the extent of oxidative stress but also by improving ion homeostasis and the hormonal balance of leaves [42]. Furthermore, the overexpression of the *γ-TMT* gene in transgenic *B. juncea* plants enhances the tolerance to abiotic stresses, such as salt, heavy metals, and osmoticums [40].

In the present study, *BnaC02.TMT.a*, an *Atγ-TMT* orthologue in *B. napus*, was isolated using nested PCR and functionally characterised. To explore the biological functions of *BnaC02.TMT.a* in rapeseed, we measured the contents of tocopherols and FAs and conducted a germination assay on sodium chloride (NaCl) stress medium in transgenic overexpression plants. We showed that the overexpression of *BnaC02.TMT.a* increased the α- and total tocopherol content, altered the proportion of FA composition, and promoted the germination rate after NaCl treatment. Under conditions involving salt-related stress, the accumulation of ROS is an important factor affecting the inhibition of seed germination, and hydrogen peroxide (H_2_O_2_) and superoxide anion (O_2_^-^) are the main components of ROS [50,51]. To obtain a deeper insight into the mechanism underlying increased salt tolerance in transgenic plants, we analysed ROS accumulation, which was estimated using diaminobenzidine (DAB) and nitroblue tetrazolium (NBT) staining and measured the H_2_O_2_ and O_2_^-^ content. Our results suggest that *BnaC02.TMT.a* prevents oxidative damage by ROS scavenging during germination in *B*. *napus* seeds under conditions involving salt stress. This study revealed that the overexpression of *BnaC02.TMT.a* could affect tocopherols, FAs, and the response to salt stress, which will be useful for engineering rapeseed with improved vitamin E content, FA composition, and salt tolerance.

## 2. Results

### 2.1. Sequence Analysis of Bnγ-TMT Paralogs

Upon performing the protein BLAST using *A. thaliana* Atγ-TMT in *Brassica napus* pan-genome information resource (BnPIR), four Bnγ-TMT paralogs were predicted in the genome of the *B. napus* cultivar “Zhongshuang11 (ZS11)”: BnaC02T0331100ZS, BnaC02T0197500ZS, BnaA02T0247300ZS, and BnaA02T0154300ZS, which were located on the chromosomes C02 and A02 and designated *BnaC02.TMT.a*, *BnaC02.TMT.b*, *BnaA02.TMT.a*, and *BnaA02.TMT.b*, respectively. The alignment of the amino acid sequences of the four Bnγ-TMT paralogs revealed high identities ranging from 96.5% to 98.8% (Appendix A). The identity of the protein sequence between the four Bnγ-TMT proteins and Atγ-TMT was at least 88.4%, whereas BnaC02.TMT.a showed a higher identity with Atγ-TMT, reaching 89.3% (Appendix A). Moreover, all Bnγ-TMT paralogs were predicted to possess four highly conserved S-adenosylmethionine (SAM)-binding domains, similar to Atγ-TMT (Figure 2). This implies that all four paralogs of Bnγ-TMT may be functionally conservative. Phylogenetic analysis was performed to investigate the evolutionary relationships between the Bnγ-TMT and 11 γ-TMT proteins in *A. thaliana* and six other oil crops. As shown in Figure 3, BnaC02.TMT.b was most closely related to Atγ-TMT, whereas BnaC02.TMT.a was closely related to BnaA02.TMT.a. Compared with γ-TMT in other species, all four paralogs were closely related to Atγ-TMT.

### 2.2. Analysis of the Expression Pattern of Bnγ-TMT

To verify the function of *Bnγ-TMT*, quantitative reverse transcription PCR (RT-qPCR) was used to explore the spatiotemporal expression pattern of all *Bnγ-TMT* paralogs in various tissues of the *B. napus* cultivar “ZS11”. As illustrated in Figure 4, *Bnγ-TMT* was widely expressed in various tissues, with higher expression levels in the flowers, leaves, and developing seeds (35 days after pollination (DAP)) than in the roots and stems. Notably, during seed development, the *Bnγ-TMT* expression remained relatively low at the embryogenesis stage from 10 to 15 DAP, decreased slightly from 20 to 26 DAP, and progressively increased at 29 DAP and 32 DAP, reaching a maximum at 35 DAP (Figure 4).

### 2.3. BnaC02.TMT.a Increased α-Tocopherol Content in B. napus Seeds

To elucidate the role of *Bnγ-TMT*, we introduced the overexpression construct *35S:BnaC02.TMT.a-6 haemagglutinin* (*6HA*) into the *B. napus* cultivar “ZS11” plants (Figure 5A). Two representative independent T_2_ homozygous lines (*OE#3* and *OE#5*) were confirmed by the PCR amplification of *BnaC02.TMT.a* using the specific primers 35S_P/*BnTMT*_R (Figure 5B). Furthermore, the expression level of the *Bnγ-TMT* gene was confirmed in these transgenic plants by RT-qPCR. The expression of *Bnγ-TMT* was significantly higher in these transgenic plants than in wild-type ZS11 (Figure 5C).

To investigate the function of *BnaC02.TMT.a* in mediating tocopherol accumulation, we measured the levels of total tocopherol and α-, β-, γ-, and δ-tocopherol in the mature seeds of wild-type ZS11 and the transgenic lines *OE#3* and *OE#5*. The levels of α-tocopherol and total tocopherol were much higher in both of the transgenic lines than those in the wild-type seeds (Figure 5D,E), whereas no apparent differences were observed in the levels of γ- and δ-tocopherol between the wild-type and transgenic plants (Appendix A). In addition, no β-tocopherol was detected in either the control or transgenic plants.

### 2.4. BnaC02.TMT.a Promoted PUFA Biosynthesis in B. napus Seeds

To elucidate the biological functions of *BnaC02.TMT.a* in seed FAs, we measured the quantities of the major FA compositions and total FAs of mature seeds between the wild-type ZS11 plants and the *BnaC02.TMT.a* homozygous transgenic lines *OE#3* and *OE#5*. In the mature seeds of the transgenic lines, a significant decrease was observed in the C18:1 (oleic acid) content along with a significant increase in the amount of C18:2 (linoleic acid) and C18:3 (linolenic acid) (Figure 6A), suggesting that α-tocopherol might affect the conversion of oleic acid to linoleic acid and linoleic acid to linolenic acid. However, no significant differences were observed with respect to the total FA content (Appendix A) and several seed morphological traits, such as the seed coat colour (Appendix A), seed size (Appendix A), and dry weight (Appendix A), between the mature seeds of the wild-type and transgenic plants.

Three critical stages of seed oil accumulation (24, 28, and 32 DAP) were carefully selected to compare the expression profiles of *BnFAD2* and *BnFAD3* between the transgenic line *OE#3* and the wild-type control. As expected, the expression of *BnFAD2* was significantly higher at the seed maturation stage from 28 and 32 DAP in the transgenic seeds than in the wild-type seeds (Figure 6B), and the expression of *BnFAD3* was consistently upregulated in the developing seeds of the *OE#3* transgenic plants at 24 and 32 DAP (Figure 6B).

### 2.5. Functional Characterisation of BnaC02.TMT.a in the Seed Germination under Conditions Involving Salt Stress in B. napus

The effects of *BnaC02.TMT.a* on seed germination under conditions involving salt stress in the wild-type and transgenic lines *OE#3* and *OE#5* were examined in sterile redistilled water with or without 180 mM NaCl. As illustrated in Figure 7 and Figure 8, *OE#3*, *OE#5*, and the control ZS11 showed similar seed germination rates in the medium without NaCl treatment. Notably, under stress conditions involving 180 mM NaCl, the differences in germination between the wild-type and transgenic seeds were apparent (Figure 7 and Figure 8). The seed germination was completely inhibited during the first 40 h after sowing (HAS) and from 40 to 192 HAS, which was significantly higher in the transgenic line than in the controls (Figure 8).

Under control conditions, the ROS levels were similar between the wild-type and transgenic lines *OE#3* and *OE#5*. In contrast, upon treatment with 180 mM NaCl, a lower DAB and NBT staining intensity, especially in cotyledons, was observed with the transgenic plants, suggesting that the ROS levels were significantly suppressed in the transgenic lines *OE#3* and *OE#5* compared to those in the wild-type ZS11 (Figure 9A). Paralleling the observations described in Figure 9A, under control conditions, no significant differences were observed in the H_2_O_2_ and O_2_^−^ contents between the wild-type and transgenic plants. Compared to the controls, the H_2_O_2_ and O_2_^−^ contents were substantially decreased in both of the transgenic plants exposed to 180 mM NaCl (Figure 9B). Therefore, our results revealed that the overexpression of *BnaC02.TMT.a* alleviated the oxidative damage to the seeds under conditions involving salt stress by ROS scavenging, thereby promoting seed germination under such conditions.

Our results suggest that *Bnγ-TMT* plays a positive role in regulating the response of *B. napus* to salt stress, which might involve complex mechanisms, such as the ROS scavenging system.

## 3. Discussion

Rapeseed is one of the most important oil supply crops and contains a significant amount of tocopherols, which are beneficial for human health. Enhancing the quality and composition of FAs and tocopherols in seeds has long been a target for rapeseed breeding. In this study, we showed that *BnaC02.TMT.a* promoted the accumulation of α- and total tocopherol, altered the FA composition in seeds, and enhanced the tolerance to salt stress during seed germination in *B. napus*.

*B. napus* is an allotetraploid species formed by the natural hybridisation between *B. rapa* and *B. oleracea* [1]. A large number of chromosome duplications, rearrangements, and deletions occurred during the evolutionary processes, resulting in an average of 2–8 paralogs of each *A. thaliana* locus in the *B. napus* genome [52]. Consistently, four paralogs to Atγ-TMT were found in the *B. napus* cultivar “ZS11” genome (Figure 2). It was shown the overexpression of *Atγ-TMT* increased the α-tocopherol concentration and vitamin E activity with unaltered total tocopherol quantity in seeds, and the *Atγ-TMT* mutation exhibited an absent α-tocopherol but high levels of γ-tocopherol accumulation in the leaves of *A. thaliana* [20,21]. We found that the overexpression of *BnaC02.TMT.a* promoted α-tocopherol accumulation in *B. napus* seeds (Figure 5D). BnaC02.TMT.a shared a high amino acid identity and a close evolutionary relationship with Atγ-TMT (Figure 2 and Figure 3). In addition, four SAM binding domains involved in methyl transfer reactions [53] were highly conserved between BnaC02.TMT.a and Atγ-TMT (Figure 2). These results suggest that BnaC02.TMT.a may exhibit a conserved role in regulating the accumulation of α-tocopherol in *A. thaliana*. In rapeseed oil, the major tocopherol form is γ-tocopherol (65%), followed by α-tocopherol (35%). Furthermore, a small proportion of δ-tocopherol (<1%), and β-tocopherol is absent [54]. *BnaA.VTE4.a1* (one *γ-TMT* paralogue from *B. napus* variety “Express”) is able to increase the α-tocopherol content because of the shift from γ- to α-tocopherol, and the total tocopherol level is not altered in transgenic *A. thaliana* seeds [55]. Interestingly, we found that the overexpression of *BnaC02.TMT*.a resulted in higher levels of α-tocopherol (Figure 5D); however, no obvious changes were observed in γ-tocopherol content in *B. napus* seeds (Appendix A), which may explain the higher levels of total tocopherol (Figure 5E). The distinct roles of *BnaA.VTE4.a1* and *BnaC02.TMT.a* in γ- and total tocopherol production might be attributed to their amino acid differences and functional divergence (Appendix A). Notably, the functional characterisation of *BnaA.VTE4.a1* was performed in a heterologous transgenic system, which may not truly reflect its function in *B. napus*. A possible reason for the unaltered γ-tocopherol content is that γ-tocopherol could be replenished by other mechanisms in the *B. napus* transgenic lines.

The levels of α-tocopherol are positively correlated with linoleic acid and PUFAs in vegetable oils [33,34]. Upon examining a collection of *B. napus* lines with low erucic acid, a significant positive correlation was found between the α-/γ-tocopherol ratio and C18:2 and PUFAs [35]. Accompanied by an increased α-tocopherol level, the FA composition was altered; that is, decreased levels of oleic acid (C18:1) and increased levels of linoleic acid (C18:2) and linolenic acid (C18:3) were observed in *35S:BnaC02.TMT.a-6HA* plants (Figure 6A). *FAD2* [56,57] and *FAD3* [58,59] are essential for the formation of C18:2 and C18:3, respectively, and higher transcript levels of *BnFAD2* and *BnFAD3* (Figure 6B) may be helpful for the accumulation of C18:2 and C18:3 in *35S:BnaC02.TMT.a-6HA* seeds. As previously reported, the association between tocopherol and FAs in oils is mostly attributed to the biochemical function of tocopherol in protecting against lipid oxidation, rendering it crucial for PUFA stability [33,34]. Furthermore, biochemical and genetic evidence indicates that tocopherols have a more limited role in photoprotection than previously assumed [45,46]. In the absence of tocopherols, the conversion of 18:2 to 18:3 in endoplasmic reticulum (ER) lipids is reduced under low temperatures, suggesting some plausible scenarios for the direct or indirect interaction of tocopherols during ER PUFA metabolism. FAD2 and FAD3 are ER-localised FA desaturases. Together with the findings indicating that *BnaC02.TMT.a* enhanced the expression of *FAD2* and *FAD3* in *35S:BnaC02.TMT.a-6HA* developing seeds (Figure 6B), we hypothesised that in addition to their antioxidant activity, the function of α-tocopherol in regulating FA formation may be attributed to the interaction between α-tocopherol and ER FA metabolism.

The role of tocopherols in abiotic stresses, such as salt [39,40,41,42,43,44], low temperature [45,46], drought [41,47], and heavy metals [40,43,48,49], has been observed. These results, together with our observations of *35S:BnaC02.TMT.a-6HA* plants under conditions involving salt stress (Figure 7, Figure 8 and Figure 9), suggest that tocopherols play a crucial role in the alleviation of salt stress. Stresses lead to excessive ROS levels causing oxidative damage [60,61]. H_2_O_2_ and O_2_^-^ are the most common forms of ROS generated during photosynthesis [50]. In this study, the overexpression of *BnaC02.TMT*.a promoted seed germination (Figure 7 and Figure 8) and reduced the production of H_2_O_2_ and O_2_^-^ in *B. napus* under conditions involving salt stress (Figure 9), indicating that *BnaC02.TMT.a* alleviated the negative effects of NaCl toxicity by scavenging ROS. This was reminiscent of the expression of *γ-TMT* in chloroplasts, accelerating the conversion of γ-tocopherol to α-tocopherol, which contributes to a decrease in susceptibility to salt [43]. Tocopherol-deficient *Arabidopsis* mutants were particularly sensitive to salt stress, highlighting the role of α-tocopherol in maintaining cellular Na^+^/K^+^ homeostasis and hormonal balance [42,47]. The overexpression of *Msγ-TMT* in alfalfa (*Medicago sativa* L.) can directly or indirectly affect the biosynthesis of multiple phytohormones that may play a comprehensive role in drought stress tolerance [47]. FAs or FA derivatives function as signalling molecules or hormones, carbon and energy storage materials, and surface layers that protect plants from environmental stresses [62,63,64]. In addition, C18:3 serves as a precursor for the biosynthesis of jasmonic acid, which is involved in the response to various stresses [65,66]. Overall, considering the changes in FA composition, especially with an increase in the proportion of C18:3 (Figure 6A), other possible reasons for the better seed germination performance of the *35S:BnaC02.TMT.a-6HA* plants in the present study might, in part, be attributed to signalling cascades connecting tocopherol levels with FAs, hormonal responses, ion homeostasis, and all aspects that warrant further investigation.

Plant growth and crop production are adversely affected by environmental stresses in nature. By uncovering the genes that underlie the tolerance adaptive trait, natural variation has the potential to be introgressed into elite cultivars [67]. Modern analytical approaches, such as genomic prediction, machine learning, and multi-trait gene editing, as well as genome–environment associations, were applied to speed up the identification and deployment of genotypic sources for climate change adaptation [67,68]. Indeed, genotypes from advanced interspecific congruity backcross exhibit promising responses to extreme conditions (i.e., heat and drought), which offers novel perspectives to breed traits of interest in the face of multiple climate changes [69]. Our study revealed that *BnaC02.TMT.a* acts as a positive regulator of tocopherol biosynthesis and salt stress response, providing a beneficial candidate for improving tocopherol activity and salt stress resistance in rapeseed breeding. Tocopherol-associated traits could be employed in traditional breeding for rapeseed improvement. Furthermore, advanced breeding strategies established in recent years have provided new avenues for harnessing tocopherols biosynthesis to improve the adaptation to stressful environments.

## 4. Materials and Methods

### 4.1. Plant Material and Growth Conditions

The *B. napus* cultivar “Zhongshuang11 (ZS11)” was used in this study. It was obtained through a material transfer agreement from the Oil Crops Research Institute of the Chinese Academy of Agricultural Sciences, Wuhan, China, and selfed for at least 10 generations. The rapeseed seeds were placed in a glass dish with three layers of filter paper soaked with sterile redistilled water and imbibed at 4 °C for one day in the dark. After imbibition, the seeds were transferred to 11 × 11 cm pots and grown in a greenhouse at 22 °C with a long day duration (LD, 16 h light/8 h dark) for six weeks. Subsequently, the plants were vernalised for four weeks at 4 °C under LD conditions in a cold chamber. After vernalisation, the plants were returned to the initial greenhouse conditions for ten weeks until harvest.

### 4.2. Protein Sequence and Phylogenetic Analysis

The protein sequences of γ-TMT were obtained from the National Center for Biotechnology Information (NCBI) database and *B. napus* pan-genome information resource (BnPIR) database (http://cbi.hzau.edu.cn/bnapus/index.php, accessed on 6 September 2022). The protein sequence alignment was performed using MUSCLE (http://www.ebi.ac.uk/Tools/msa/muscle/, accessed on 8 September 2022). The S-adenosyl methionine-binding domain was predicted using the Conserved Domain Search program (http://www.ncbi.nlm.nih.gov/Structure/cdd/wrpsb.cgi, accessed on 8 September 2022). The phylogenetic analysis was performed using MEGA7 to confirm the evolutionary relationships between the γ-TMT protein sequences. A bootstrap analysis with 1000 replicates was performed to assess the statistical reliability of the tree topology.

### 4.3. Gene Cloning and Plasmid Construction

Owing to a high sequence similarity among the four paralogs of *Bnγ-TMT*, nested PCR was used for cloning *BnaC02.TMT.a*. Two pairs of PCR primers (specific and nested primers) were designed. The specific primers were designed based on the alignment of four Bnγ-TMT paralogs, and the nested primers were designed based on the full-length coding sequence (CDS) of *BnaC02.TMT.a* (BnaC02T0331100ZS) without a stop codon derived from the BnPIR database. To obtain full-length *BnaC02.TMT.a* cDNA, the total RNA was extracted from the developing seeds of the *B. napus* cultivar “ZS11” using the SteadyPure Plant RNA Extraction Kit (Accurate Biology, Changsha, China) and was subjected to reverse transcription to obtain first-strand cDNA according to the manufacturer’s instructions (TransGen, Beijing, China).

The full-length BnaC02.TMT.a CDS was amplified by nested PCR using specific and nested primers. The first pair of the PCR primers amplified a specific fragment using the primers BnaC02.TMT.a_Fc and BnaC02.TMT.a_Rc. The second pair of primers, called nested primers, was used to clone the full-length CDS of BnaC02.TMT.a. The sequence was ligated into the binary vector pGreen-35S-6HA under the control of the CaMV35S (35S) promoter, which had been digested by Xho1 and Cla1 [70]. The primers used in this study are listed in Appendix A.

### 4.4. Generation of B. napus Transgenic Plants

To obtain the transgenic *B. napus* plants, the hypocotyl segments of the *B. napus* cultivar “ZS11” were infected with the plasmid *35S:BnaC02.TMT.a-6HA* by *Agrobacterium tumefaciens* strain GV3101 [71]. Transgenic T_0_ plants were selected on MS agar medium containing 10 μg/mL phosphinothricin, and a successful transformation was confirmed by DNA genotyping according to the PCR amplification performed using gene-specific primers. The transformants were selected by genotyping according to PCR amplification, and homozygous seeds from the T_2_ generation were used for subsequent experiments after cultivation under the same aforementioned conditions.

### 4.5. Morphological Observation of Mature Seeds

The T_3_ plants of the *BnaC02.TMT.a* homozygous transgenic lines (*OE#3* and *OE#5*) and wild-type ZS11 controls were grown in a greenhouse under the aforementioned conditions. Siliques were harvested from the middle and basal parts of the main inflorescence. Subsequently, mature seeds were randomly selected from the different lines and observed using a stereomicroscope (Olympus SZ 61, Tokyo, Japan).

### 4.6. Measurement of the FA Content

The T_3_ plants of the *BnaC02.TMT.a* homozygous transgenic lines (*OE#3* and *OE#5*) and wild-type ZS11 controls were grown in a greenhouse under the aforementioned conditions. For the FA measurement, seeds were harvested from the middle and basal parts of the main stem of eight individual plants from each line. The FAs were extracted and analysed as previously described [72,73], with minor modifications. First, the seeds were ground in a mortar, and approximately 8 mg of seed powder was heated at 80 °C in a methanol solution containing 2.5% (*v*/*v*) H_2_SO_4_ for 2 h, after which the total FAs were converted to FA methyl-esters. After cooling to room temperature, the FA methyl-esters were extracted with 1 mL hexane and 2 mL 0.9% (*w*/*v*) NaCl, and the organic phase was transferred to autoinjector vials. The gas chromatography (GC-2010 Plus, Shimadzu, Japan) analysis was performed using methyl heptadecanoate as an internal standard. The concentration of each FA species was normalised to that of the internal control.

### 4.7. HPLC Analysis of the Tocopherol Content

The tocopherol analysis was performed by Suzhou Michy Biology (http://www.michybio.com, accessed on 8 July 2021). The dried seeds from eight individual plants (transgenic lines and wild-type ZS11) were sent to the company. Briefly, 30–80 mg of mature seeds were ground, and the powder was mixed with 1 mL of ethyl alcohol extraction buffer containing 1% butylated hydroxytoluene (BHT) and 1 mL of 100 g/L potassium hydroxide solution. After incubation for 40 min (85 °C) and cooling to room temperature, 2 mL of petroleum ether was added. After mixing and centrifugation, the lower organic layer was recovered, dried under nitrogen, and resuspended in 0.4 mL methanol for the HPLC analysis.

The tocopherol content of the organic extract was determined using an Agilent 1100 HPLC system. A total of 10 μL of the sample was chromatographically analysed using a Compass C18 column (250 × 4.6 mm length, 5 μm particle size) and a solvent system consisting of methyl alcohol as the mobile phase with a flow rate of 1 mL/min. The column temperature was 20 °C, and the sample was scanned at 294 nm. The identification and quantification of tocopherols were performed by comparing the retention time and peak area to tocopherol standards (Appendix A).

### 4.8. ROS Staining

As previously described [74], the seeds collected at 96 h after sowing (HAS) were incubated in the dark at 37 °C. The seeds were saturated with 0.1% (*w*/*v*) diaminobenzidine (DAB, Sigma-Aldrich, St. Louis, MO, USA) in 10 mM 2-(*N*-morpholino) ethanesulphonic acid (MES, pH 6.5) for 8 h for the H_2_O_2_ staining and with 0.1% (*w*/*v*) nitroblue tetrazolium (NBT, Sigma-Aldrich, St. Louis, MO, USA) in 50 mM potassium phosphate buffer (pH 6.4) for 10 min for superoxide staining. Finally, the seeds were destained with 95% (*v*/*v*) ethanol.

### 4.9. ROS Quantification

The H_2_O_2_ content was measured using the titanium–peroxide complex method, as described previously, with minor modifications [74,75,76]. A total of 0.1 g (fresh weight) of the seeds collected at 96 HAS were ground in liquid nitrogen and extracted with 1 mL of cooled acetone. After the vortex mixing, the samples were centrifuged for 10 min at 8,000× g at 4 °C. The supernatant was then added to a solution comprising 0.1 mL of 5% (*w*/*v*) titanium sulphate and 0.2 mL of ammonia. The mixture was centrifuged for 10 min at 8,000× g. After removing the supernatant, the precipitate was dissolved in 1 mL of 2 M H_2_SO_4_. The absorbance was immediately measured at 415 nm. The H_2_O_2_ content was calculated using a standard curve generated using known concentrations of H_2_O_2_.

The O_2_^−^ content was quantified as previously described, with minor modifications [74,76]. First, 0.1 g (fresh weight) of the seeds at 96 HAS was ground in liquid nitrogen and added to 1 mL of K-phosphate buffer (65 mM, pH 7.8). The homogenates were centrifuged at 12,000× *g* for 20 min (4 °C). A total of 1 mL 0.1 M hydroxylamine was added to the supernatant, and the mixture was incubated at 25 °C for 20 min. Then, 1 mL of 7 mM α-naphthalenamine and 1 mL of 58 mM 4-aminobenzenesulfonic acid were added, and the samples were centrifuged for 20 min at 12,000× *g* at 4°C. Trichloromethane (1 mL) was added to eliminate the pigments, and the mixture was centrifuged at 10,000× *g* for 5 min. The absorbance of the supernatant was measured at 530 nm. The O_2_^-^ content was determined using a standard curve generated from known concentrations of NaNO_2_.

### 4.10. Germination Test

For the seed germination assay, 50 plump seeds from each treatment group were carefully picked and washed with sterile redistilled water thrice, evenly placed on a glass dish with three layers of filter paper soaked in the same volume of 180 mM NaCl solution or sterile redistilled water as the control, and imbibed at 4 °C for one day in the dark. After imbibition, the seeds were transferred to a growth chamber for germination. Every other day, the germinating seeds were transferred to a new glass dish with three layers of filter paper soaked in the same volume of 180 mM NaCl solution or sterile redistilled water until the seed germination experiment was completed. Three independent biological replicates were obtained for the control and salt treatments of the transgenic lines (*OE#3* and *OE#5*) and ZS11. Seed germination was defined as the complete emergence of the radicle through the seed coat [77]. The number of germinated seeds was calculated every 8 h until the completion of the experiment.

### 4.11. Gene Expression Analysis

The roots, stems, flowers, and developing seeds for the spatial and temporal expression analysis were harvested from at least eight individual plants grown in different pots. The total RNA was isolated using the SteadyPure Plant RNA Extraction Kit (Accurate Biology, Changsha, China) and reverse-transcribed using EasyScript One-Step gDNA Removal and cDNA Synthesis SuperMix (TransGen, Beijing, China) according to the manufacturer’s instructions. Quantitative reverse transcription RT-PCR (RT-qPCR) was performed in 96-well blocks using SYBR Green Master Mix (Cofitt, Hong Kong, China) with three biological replicates and three technical replicates. Considering the high sequence similarity among the paralogs of *Bnγ-TMT*, the primers were designed for the conserved regions within groups of paralogs to detect joint gene expression. The relative expression levels were calculated using a modified double delta method [78]. The housekeeping gene encoding glyceraldehyde-3-phosphate dehydrogenase (BnGAPDH), a key enzyme in the process of glycolysis and gluconeogenesis, was considered as a reference, which is widely founds in various organisms and is regarded as an internal control [79]. The primers used for RT-qPCR are listed in Appendix A.

### 4.12. Statistical Analysis

A completely randomised design was used in this study. All data were classified using Win-Excel and reported as the mean ± standard deviation (SD). The data were analysed by one-way analysis of variance (ANOVA) using SPSS software (version 17.0, SPSS Inc., Chicago, IL, USA). Significant differences were determined by Student’s *t*-test (*p* ≤ 0.05).

## 5. Conclusions

In summary, we discovered new functions for *Bnγ-TMT* in the regulation of tocopherol and FA biosynthesis and salt tolerance in rapeseed. Since it is an allopolyploid species, four paralogs of *Bnγ-TMT* were found in the *B. napus* genome. The protein sequence and phylogenetic analyses suggested that all four Bnγ-TMT paralogs might exhibit functions similar to those shown by Atγ-TMT for determining tocopherol accumulation. Together with the expression data, these results further suggest that *Bnγ-TMT* might regulate tocopherol accumulation mainly during the maturation stage in *B. napus* seeds. Overexpression of *BnaC02.TMT.a* promotes seed α- and total tocopherol accumulation during seed maturation. Additionally, the fatty acid composition was altered. Our findings suggest that *BnaC02.TMT.a* promoted the conversion of oleic acid to linoleic acid and subsequently linoleic acid to linolenic acid by upregulating the expression of *BnFAD2* and *BnFAD3* in rapeseed. Furthermore, *BnaC02.TMT.a* plays a crucial role in mediating the response to salt stress during seed germination, which involves complex mechanisms, including ROS scavenging capability, signalling cascades connecting tocopherol with FAs, hormonal responses, and ion homeostasis. This study broadens our understanding of the function of the *Bnγ-TMT* gene and provides a novel strategy for genetic engineering to improve rapeseed breeding.

## Figures and Tables

**Figure 1 ijms-23-15933-f001:**
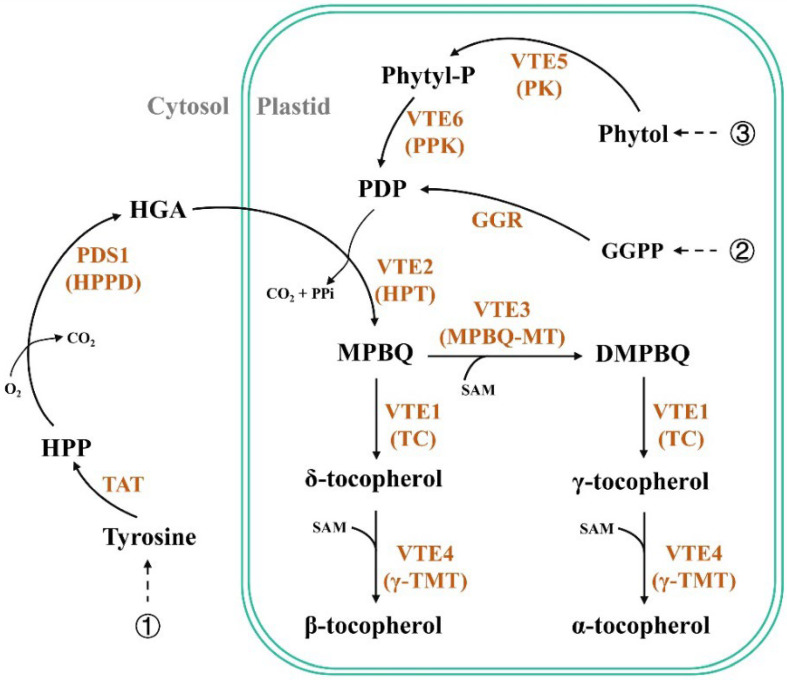
Simplified schematic diagram showing the tocopherols biosynthetic pathway in plants: ① shikimic pathway; ② methylerythritol phosphate pathway; ③ phytol recycling pathway. TAT: tyrosine aminotransferase; HPP: *p*-hydroxyphenylpyruvic acid; PDS1 (HPPD): HPP dioxygenase; HGA: homogentisic acid; VTE5 (PK): phytol kinase; Phytyl-P: phytyl phosphate; VTE6 (PPK): phytyl phosphate kinase; GGPP: geranylgeranyl diphosphate; GGR: geranylgeranyl reductase; PDP: phytyl diphosphate; VTE2 (HPT): HGA phytyl transferase; MPBQ: 2-methyl-6-phytyl-1,4-benzoquinol; VTE3 (MPBQ-MT): MPBQ methyltransferase; SAM: S-adenosylmethionine; DMBPQ: 2,3-dimethyl-6-phytyl-1,4-benzoquinol; VTE1 (TC): tocopherol cyclase; VTE4 (γ-TMT): γ-tocopherol methyltransferase.

**Figure 2 ijms-23-15933-f002:**
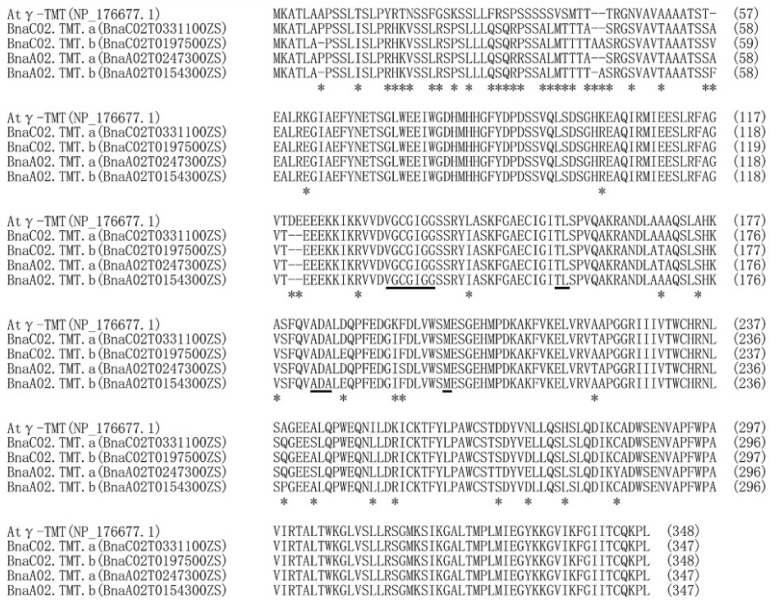
Alignment of amino acid sequences of γ-TMT derived from *Arabidopsis thaliana* and *Brassica napus*. The alignment was conducted using the MUSCLE program (http://www.ebi.ac.uk/Tools/msa/muscle/, accessed on 8 September 2022), and the different amino acids are indicated by asterisks. S-adenosylmethionine binding sites are underlined.

**Figure 3 ijms-23-15933-f003:**
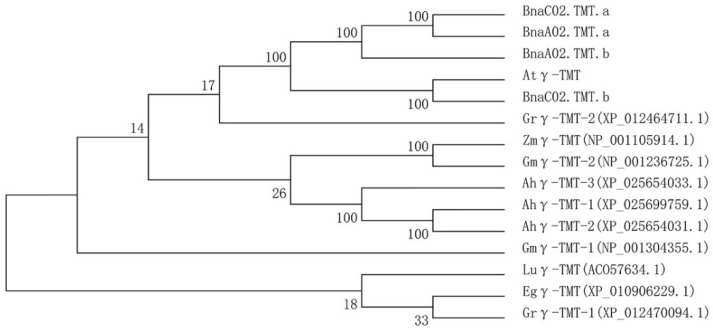
Phylogenetic tree analysis of γ-TMT proteins derived from *A. thaliana* and seven oil crops. A neighbour-joining tree (Jones–Taylor–Thornton model) was generated by MEGA7. A bootstrap analysis with 1000 replicates was performed to assess the statistical reliability of the tree topology. The accession numbers corresponding to the species names are listed as follows: *A. thaliana*, Atγ-TMT (NP_176677.1); *B. napus*, BnaC02.TMT.a (BnaC02T0331100ZS), BnaC02.TMT.b (BnaC02T0197500ZS), BnaA02.TMT.a (BnaA02T0247300ZS), and BnaA02.TMT.b (BnaA02T0154300ZS); *Zea mays*, Zmγ-TMT (NP_001105914.1); *Elaeis guineensis*, Egγ-TMT (XP_010906229.1); *Linum usitatissimum*, Luγ-TMT (ACO57634.1); *Glycine max*, Gmγ-TMT-1 (NP_001304355.1) and Gmγ-TMT-2 (NP_001236725.1); *Gossypium raimondii*, Grγ-TMT-1 (XP_012470094.1) and Grγ-TMT-2 (XP_012464711.1); *Arachis hypogaea*, Ahγ-TMT-1 (XP_025699759.1), Ahγ-TMT-2 (XP_025654031.1), and Ahγ-TMT-3 (XP_025654033.1).

**Figure 4 ijms-23-15933-f004:**
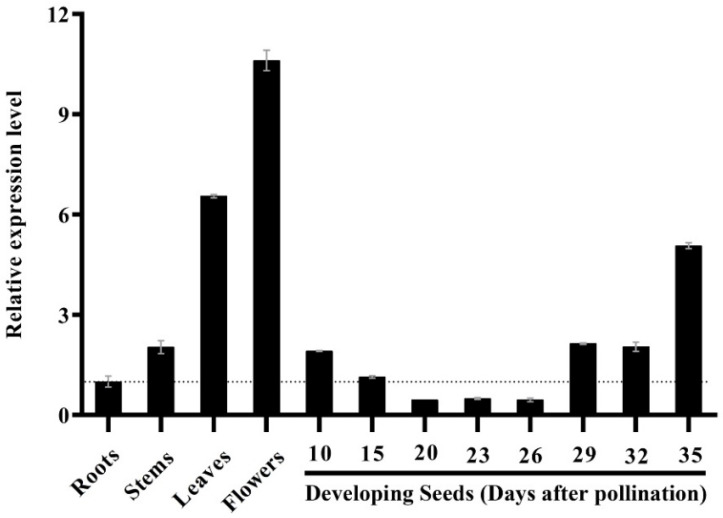
RT-qPCR analysis of the *Bnγ-TMT* expression in various tissues and developing seeds at different developmental stages in the *B. napus* cultivar “ZS11”. The RT-qPCR results were normalised against *BnGAPDH* expression as an internal control, and the expression level of *Bnγ-TMT* in roots was set to 1. The values indicate the means of three replicates of dilutions of cDNA obtained from three independent RNA extractions. The error bars denote standard deviations.

**Figure 5 ijms-23-15933-f005:**
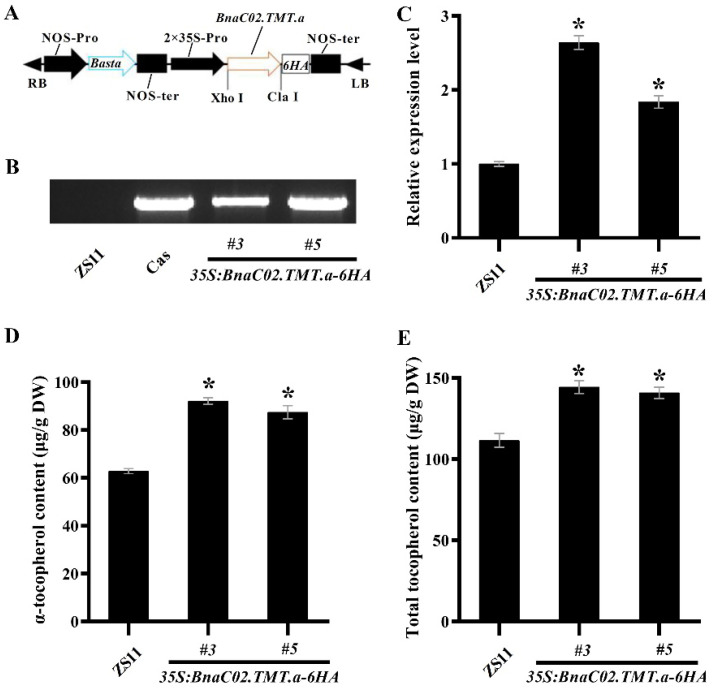
Characterisation of *35S:BnaC02.TMT.a-6HA* lines. (**A**) Schematic diagram of the constitutive expression cassette of the *35S:BnaC02.TMT.a-6HA* gene in the binary vector pGreen 2 × 35S used for plant transformation. RB, right border; LB, left border; NOS-pro, nopaline synthase promoter; NOS-ter, nopaline synthase terminator; Basta, glyphosate; 35S-pro, CaMV 35S promoter. (**B**) PCR-based genotyping of the *35S:BnaC02.TMT.a-6HA* transgenic plants using specific primers for the 35S_P/*BnTMT*_R. Cas, cassette. (**C**) Expression analysis of *Bnγ-TMT* in the *35S:BnaC02.TMT.a-6HA* transgenic plants using RT-qPCR. The RT-qPCR results were normalised against the expression of *BnGAPDH*, which was used as an internal control. The values are the means ± SD (n = 3). (**D**,**E**) The levels of α-tocopherol and total tocopherol in the mature seeds of the transgenic lines and control plants (ZS11). The asterisks denote statistically significant differences between the wild-type ZS11 and the *35S:BnaC02.TMT.a-6HA* transgenic plants (Student’s *t-*test, *p* ≤ 0.05). The error bars denote the standard deviation.

**Figure 6 ijms-23-15933-f006:**
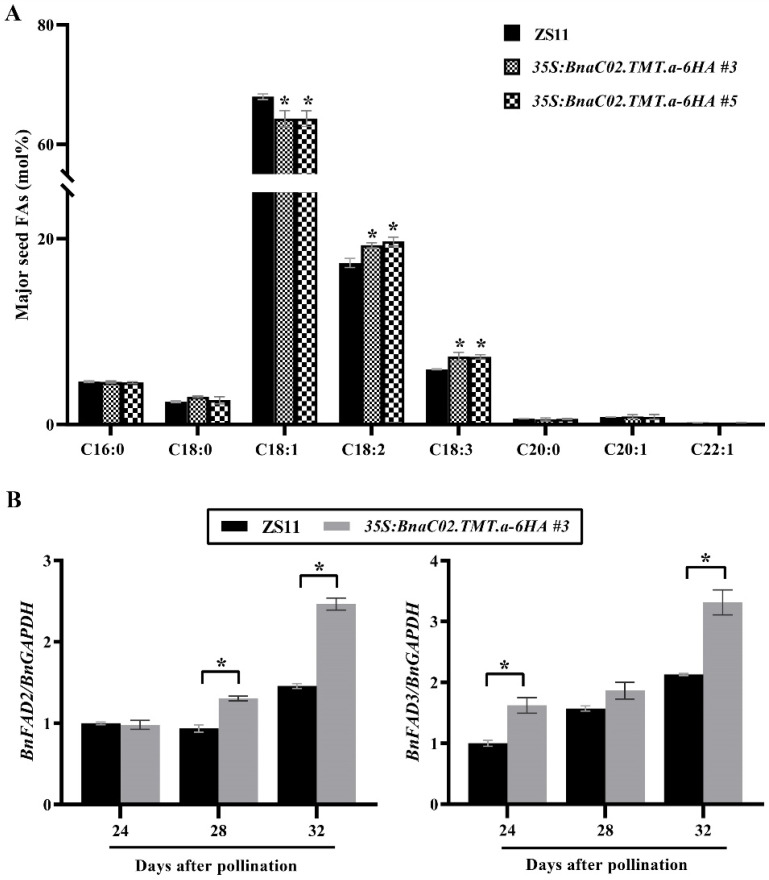
Effect of *BnaC02.TMT.a* on FA metabolism in seeds. (**A**) Comparison of the FA compositions of major seeds between the wild-type ZS11 and *35S:BnaC02.TMT.a-6HA* plants. The asterisks indicate significant differences in the total FA levels in the seeds (two-tailed paired Student’s *t*-test, *p* ≤ 0.05). The error bars denote the standard deviation. (**B**) Comparison of the expression of genes contributing to the FA modification in the developing seeds of the ZS11 and *BnaC02.TMT.a-6HA* plants. The asterisks indicate significant differences between the wild-type ZS11 and *35S:BnaC02.TMT.a-6HA* plants (two-tailed paired Student’s *t*-test, *p* ≤ 0.05). The error bars denote the standard deviation.

**Figure 7 ijms-23-15933-f007:**
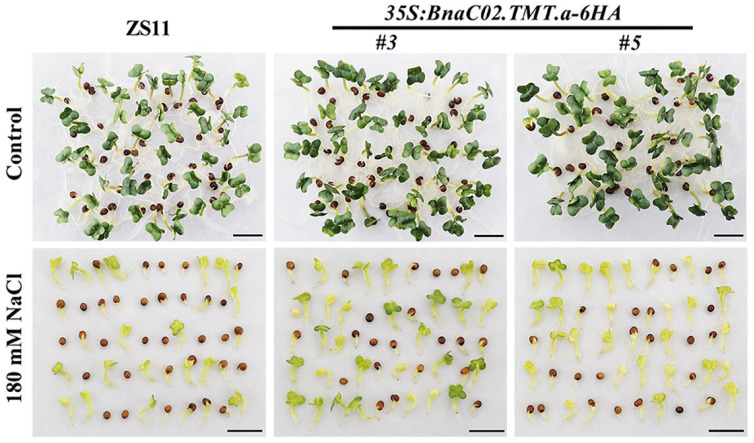
Phenotypes of the seed germination at 96 h after sowing (HAS) in the wild-type ZS11 and *35S:BnaC02.TMT.a-6HA* plants under control conditions and after treatment with 180 mM NaCl. The emergence of a visible radicle was used as a morphological marker for seed germination. Bar = 1 cm.

**Figure 8 ijms-23-15933-f008:**
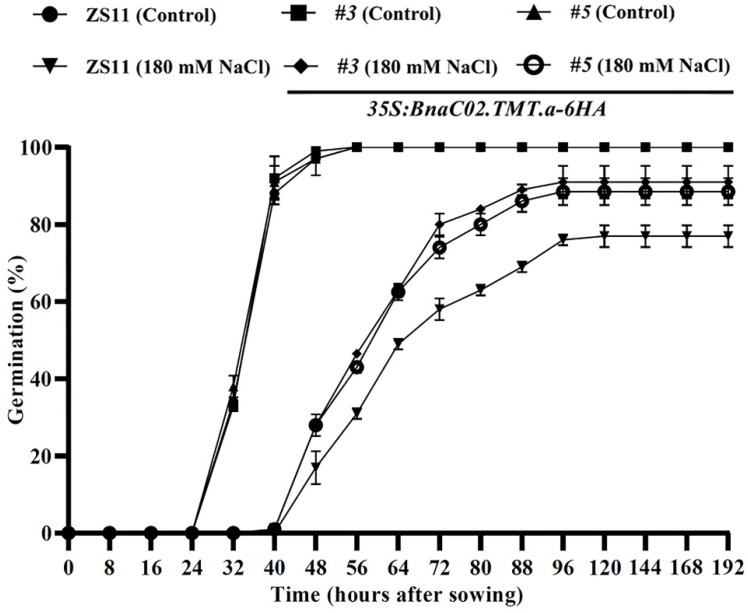
Time course analysis of seed germination in the wild-type ZS11 and *35S:BnaC02.TMT.a-6HA* plants under control conditions and after treatment with 180 mM NaCl. Seed germination was scored every eight hours, and the emergence of a visible radicle was used as a morphological marker. The data are expressed as the mean ± SD from three independent experiments evaluating 50 individual seeds.

**Figure 9 ijms-23-15933-f009:**
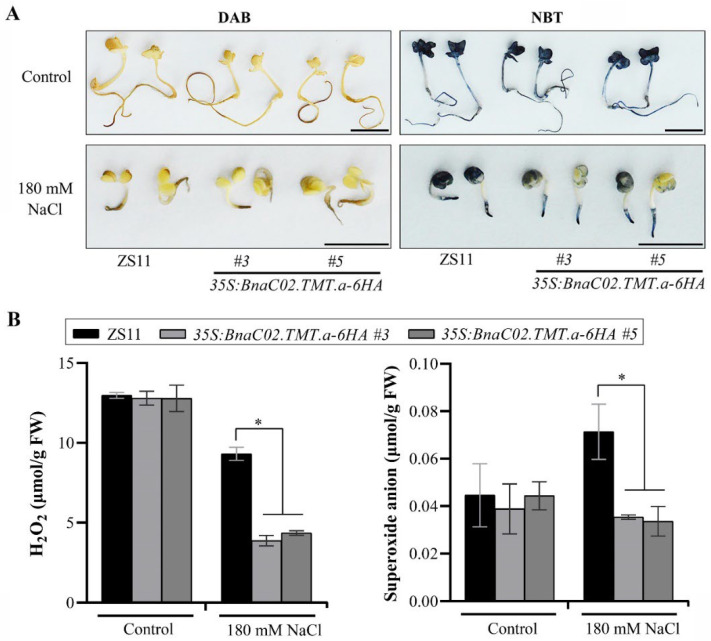
Effect of *BnaC02.TMT.a* on ROS accumulation. (**A**) diaminobenzidine (DAB, Left) and nitroblue tetrazolium (NBT, right) staining of ROS in the wild-type ZS11 and *35S:BnaC02.TMT.a-6HA* plants treated with 180 mM NaCl at 96 h after sowing (HAS). Bar = 1 cm; (**B**) measurements of H_2_O_2_ (Left) and O_2_^−^ (right) in the wild-type ZS11 and *35S:BnaC02.TMT.a-6HA* plants in response to treatment with 180 mM NaCl at 96 HAS. The asterisks indicate significant differences between the wild-type ZS11 and *35S:BnaC02.TMT.a-6HA* plants under control conditions or after 180 mM NaCl treatment, respectively (two-tailed paired Student’s *t*-test, *p* ≤ 0.05). The error bars denote the standard deviation. FW, fresh weight.

## Data Availability

All data included in this study are available upon reasonable request by contact with the corresponding author.

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
