# Peer review of "Effect of Overexpression of γ-Tocopherol Methyltransferase on α-Tocopherol and Fatty Acid Accumulation and Tolerance to Salt Stress during Seed Germination in Brassica napus L."

_ijms, 2022, doi:10.3390/ijms232415933_

Round 1

Reviewer 1 Report

The manuscript: „The effect of BnaC02.TMT.a on the a-tocopherol and fatty acid accumulation, and seed germination under salt stress in Brassica napus L.” presents a study on potential improvement of the seed oil quality by overexpression of g-tocopherol-methyltransferase (g-TMT) in B. napus cultivar ‘Zhongshuang11’. Moreover, the Authors showed enhanced seed germination under salt stress in transformed plants.

The topic of this work refers to an important theoretical problem of broad interest, concerning improving the quality of seed oil in oil crops regarding its nutritional value and importance for human health by increasing the content of a-tocopherol in total seed oil tocopherols content.

Resulting from protein sequence analyses from the NCBI and BnPIR databases, the BnaC02.TMT.a gene, one of the four paralogs of the B. napus g-TMT gene was chosen by the Authors for further analyses. Specific primers were designed and used for gene cloning under the CaMV35S promoter. Transgenic homozygous plants were obtained and analyzed with respect to control, wild-type plants. Reverse transcription quantitavive PCR, RT-qPCR, was used to check for relative expression of the BnaC02.TMT.a gene, using the BnGADPH houskeeping gene as an internal control, in the transformed- and control plants. Total FA content in addition to both, FA and tocopherols composition were measured in seed oil of the transformed and control plants. Also, seed germination was analyzed under salt stress in addition to ROS accumulation in transgenic and control plants.

The obtained results led to conclusion that overexpression of the BnaC02.TMT.a gene led to an increase in a-tocopherol content, and at the same time - to significant decrease in oleic- and a significant increase of linoleic- and linolenic acids. Moreover, the transformants were tolerant to 180 mM NaCl salt stress, as compared to control, wild-type plants.

In my opinion, the manuscript comprises a valuable study and results, and could be accepted after minor revisions, as follows:

The title

Preferably, the title could be changed, for example: “The effect of overexpression of g-tocopherol-methyltransferase on a-tocopherol and fatty acid accumulation, and seed germination under salt stress in Brassica napus L.”

Abstract

Abstract should be re-written, so that the reader could clearly recognize the background/ hypothesis, the aim of the study, the plant material and methods used, and the main results, conclusion and further prospects.

Introduction

Line 29, ‘AACC’ should be explained;

l. 31, and vitamin E == content (?); also, it should be stated that vitamin E is present only in the cold-pressed oil;

l. 31, major FAs present, instead of ‘found’;

l. 39, reference(s) needed;

l. 40, “digging out” sounds not much scientific…;

l. 41 ‘germplasms’, instead of: ‘germplasm’;

l. 43, ‘consisting’, instead of: ‘mainly consists’;

lanes: 49-66, the description of biochemical pathways involved in tocopherols’ biosynthesis should be more consistent, and clear. Moreover, a schematic diagram should be included.

l. 57, current knowledge on the g-TMT function in A. thaliana and crop plants should be presented in a systematic, comprehensive way. Also, it should be stated that B. juncea L. has an ‘AA’ genome. In addition to the Latin binomial nomenclature, common names of crop species should be included.

The aim of the study should be clearly stated in the Introduction.

Results

l.77, ‘C02, A02’, instead of: ‘Co2, C02, A02, A02’;

l.82, data in the Figure S1 seem to be not necessarily adequate to those included in the “Results”, they should be verified again;

l. 90-92, should be moved to “Conclusions”

Figure 1. - I would suggest to divide this figure, and instead of: ‘Figure 1., A and B’, it should be ‘Figure 1.’, and then ‘Figure 2.’;

l. 94 ‘aminoacid sequences’, instead of: ‘protein sequences’;

l. 105, it looks like: RT-qPCR, ‘quantitative reverse transcription PCR’, instead of: ‘quantitative real-time `PCR’;

l. 117-118, it is ‘a conclusion’ and not ‘the result’;

l. 140-141, it looks like ‘conclusion’, and not ‘the result’;

l. 155-157, I would suggest to move this fragment to the “Introduction”

l. 176-177, to: “Discussion”;

l. 183-190, should be moved to “Conclusions”;

l.194-197, to: “Introduction”

l.204/205, to: “Conclusions”

Figure 5. A, and B – should be divided into two figures, and the pictures should be larger, as in the present form, they are unclear;

l. 214-219, to: “Introduction”;

Discussion

l. 242, “rapeseed serves as major sources” (??);

l. 252, and other – ‘it is known’, or, ‘it was shown’, instead of: ‘previous studies reported’;

Materials and Methods

l. 346-351, obtaining of transformed plants should be described in more details;

l. 434, BnGAPDH, the name of the enzyme should be explained; why was that gene chosen as an internal control?

In addition, the whole manuscript should be verified with respect to keep it clear, simple and coherent.

Author Response

Dear Reviewer 1,

We have taken all comments from you into consideration, and made a thorough revision on this manuscript. I am sending you the thoroughly revised version of this manuscript for your reviewing, and hope that the revised version will meet the requirement of International Journal of Molecular Sciences.

Please find the responses to your comments in Word file, thanks a lot! 

Sincerely,

Mingxun Chen and Yuan Guo

Reviewer 2 Report

The manuscript titled "The effect of BnaC02.TMT.a on the α-tocopherol and fatty acid 2 accumulation, and seed germination under salt stress in Bras- 3 sica napus L." has interesting findings regarding the functional characterization of  BnaC02.TMT.a after overexpressing in B. napus. However, certain major modifications may improve the quality of the manuscript and could be worth publishing.

1. The title represent that functional role of BnaC02.TMT.a may affect tocopherol, seed germiantion and other parameters under salt stress, however, only seed gemination, ROS have been anaylsed under salt stress, this manuscript does not give any result about the change in totpherol in transgenic B. napus under salt stress, which is a serious concern.

2. Introduction section does not show any literature regarding salt stress and its alleviation by transgenic lines. 

3. Line 12-13 Desirable tocopherols and fatty acids (FAs) in 12 rapeseed are interesting of breeding due to their --- does not show proper meaning. the word interesting does not suit gramatically. Moreover,there are serious grammatical mistakes, therefore the whole document need attention and revision.

4. Figure 2.which house keeping gene was used before performing qRT-PCR?

5. primers sequence is not given in the supplementary file.

6. in line 132, it is significantly higher instead of significant higher.

7. Primer sequence of 2*35 -Pro and NoS terminal are not given.

8. In Figure 5, Although the graph shows clear difference in germination rate however the image does not show that.

9. What is the corelation of analysing the effect of overexpression of BnaCO2.TMT.a on tocpherol and germinatio uder salt stress?

10. Figure 6 A does not show clear differnce in H2O2 as the DAB color is same in both controlled and transgenic lines.

11. Discussion is just a dictation, improve it by justifying your result with related work. 

Follow proper journal format.

Author Response

Dear Reviewer 2,

We have taken all comments from you into consideration, and made a thorough revision on this manuscript. I am sending you the thoroughly revised version of this manuscript for your reviewing, and hope that the revised version will meet the requirement of International Journal of Molecular Sciences.

Please find the responses to your comments in Word file, thanks a lot! 

Sincerely,

Mingxun Chen and Yuan Guo

Reviewer 3 Report

In this study, the authors have focused on evaluating the effects of BnaC02.TMT.a on the alpha-tocopherol and fatty acid accumulation and seed germination under salt stress in Brassica napus L. The results attained are of interest and have had scientific facets. However, the current form of paper is not well documented and organized. There are many flaws and unclear methods described as well as poorly English written as an acceptable scientific paper. Hence, the authors are requested to totally revise and edit, some comments and suggestions may be useful to improve the quality of the paper below (further see in pdf file).

1. Abstract should be reworded and edited in order to reduce the background and context information, the abstract part is not enough place for such information (see in pdf file)

2. Introduction must be rewritten, should mention the salt stress and explain why it is important to do this study. At the end of the introduction part, the specific objectives of this study should be narrated.

3. Fig 1 should be replaced and provided a high-quality and solution version. The value of error bars in all Figs must be added

4. The materials and methods must be totally revised.  Much important information needs to provide. For example, How long did the seeds grow in the growth chamber? In gene cloning and plasmid construction, this section is unclear protocol and without any cited references. It should be reworded in detail. 

5. In subsection 4.4, generation of B napus transgenic plants, Did you have any genotyping data to confirm successful transformation?

6. In 4.5, where did the plants cultivate to harvest the siliques?

7. In 4.6, where did the plants cultivate to harvest the siliques?

8. In 4.10, the germination test, this method must be rewritten and added some information in detail, for example, what concentration of NaCL, and time of treatment etc.

9. Numerous flaws and confusing sentences as well as grammar errors are available and must be judiciously revised and edited.

Author Response

Dear Reviewer 3,

We have taken all comments from you into consideration, and made a thorough revision on this manuscript. I am sending you the thoroughly revised version of this manuscript for your reviewing, and hope that the revised version will meet the requirement of International Journal of Molecular Sciences.

Please find the responses to your comments in Word file, thanks a lot! 

Sincerely,

Mingxun Chen and Yuan Guo

Round 2

Reviewer 2 Report

The author has tried to justify all questions still the manuscript has become a mess. 

I could not find any statistical tool used to analyze the result.

Which salt stress marker gene was used to determine whether the stress was adequately applied?

NBT and DAB still not clear.

Author Response

Dear Reviewer 2,

Thank you very much for your time and efforts in reviewing our manuscript. We are very grateful to you for reviewing the manuscript so carefully. I am sending you the thoroughly revised version of this manuscript for your reviewing, and hope that the revised version will meet the requirement of International Journal of Molecular Sciences.

Please find the responses to your comments in Word file, thanks a lot! 

Sincerely,

Mingxun Chen and Yuan Guo
